# Septic Hyperinflammation—Is There a Role for Extracorporeal Blood Purification Techniques?

**DOI:** 10.3390/ijms25063120

**Published:** 2024-03-08

**Authors:** Dominik Jarczak, Stefan Kluge, Axel Nierhaus

**Affiliations:** Department of Intensive Care Medicine, University Medical Center Hamburg-Eppendorf, 20251 Hamburg, Germany; d.jarczak@uke.de (D.J.); s.kluge@uke.de (S.K.)

**Keywords:** sepsis, septic hyperinflammation, blood purification, immune response, cytokines, endotoxin, hemodynamics, extracorporeal therapy

## Abstract

This manuscript investigates the role of extracorporeal blood purification techniques in managing septic hyperinflammation, a critical aspect of sepsis characterized by an uncontrolled immune response leading to multiorgan dysfunction. We provide an overview of sepsis, focusing on the dynamics of immune response, the involvement of neutrophils, and the role of the endothelium in the disease’s progression. It evaluates the effectiveness of various blood purification methods, including high-cut-off membranes, high-volume hemofiltration, adsorption techniques, and albumin dialysis, in removing cytokines and endotoxin and improving hemodynamic stability. Despite some very promising results, we conclude that the current evidence does not strongly support these techniques in significantly improving survival rates in septic patients, clearly underlining the need for further research.

## 1. Introduction

### 1.1. What Is Septic Hyperinflammation, and Why Should It Be Treated?

Sepsis is a life-threatening clinical condition with extensive physiological and biochemical abnormalities. Each year, approximately 49 million people worldwide are affected by sepsis, and it is estimated that 11 million deaths can be attributed to this syndrome. This accounts for up to 19.7% of all global deaths [1]. Although there appears to be a global decline in the average mortality rate, the current mortality rate for sepsis can still reach up to 25%. In the case of septic shock, which is a subset of sepsis characterized by profound circulatory, cellular, and metabolic disturbances, the hospital mortality rate approaches almost 60% [2].

Over recent decades, the definition of “sepsis” has continuously evolved, adapting to the expanding scope of knowledge. The current definition, established by the Third International Consensus (Sepsis-3), characterizes sepsis as “organ dysfunction caused by a dysregulated host response to infection” [3]. This definition notably emphasizes, for the first time, the critical role of both the innate and adaptive immune responses in the development of the clinical syndrome. Sepsis, unlike an uncomplicated and localized infection, involves a complex disruption of the finely tuned balance between pro- and anti-inflammatory processes. Although understanding of the development, pathophysiology, and immunological mechanisms of sepsis has advanced significantly over the last three decades, the syndrome’s complexity—with its myriad interactions and effects on various organs—means that the opportunities for successful and specific therapeutic interventions remain limited.

Even approaches within the realm of personalized or “precision medicine”, where treatments are tailored to predefined conditions or the specific needs of individual patients, have yet to achieve widespread success. As our understanding of the numerous processes and interactions in sepsis expands, it becomes increasingly apparent that there might be an optimal timing for enhancing or suppressing each element of the immune response in the fight against severe infections. The demands placed on the immune system at the onset of sepsis, when the pathogen load is particularly high, differ markedly from those at later stages, when effective anti-infective treatments have usually succeeded in substantially reducing the pathogens load. In other words, administering a treatment that is theoretically appropriate at the wrong point in time can potentially worsen clinical outcomes [4]. The timing of correct diagnosis and the initiation of suitable causal, supportive, and adjunctive measures are, therefore, critical factors. Consequently, increasing global awareness of sepsis and promoting quality improvement initiatives in this field, along with the development of novel diagnostics and interventions, are essential to effectively enhance patient survival [5].

Sepsis necessitates timely and effective treatment strategies that vary across its continuum, from early sepsis to sepsis syndrome/severe sepsis and septic shock. The treatment modalities are multidisciplinary and escalate in intensity with the progression of the disease. Below is a concise overview of treatment strategies across the stages of sepsis, with a focus on when blood purification techniques could potentially be introduced.

Early Sepsis:Antimicrobial therapy: Prompt initiation of broad-spectrum antibiotics, tailored based on suspected sources of infection and local microbial resistance patterns.Source control: Identification and management of the infectious source, such as drainage of abscesses or removal of infected devices.Fluid resuscitation: Administration of intravenous fluids to restore hemodynamic stability.Supportive care: Oxygen supplementation and use of vasopressors if necessary to maintain adequate blood pressure.

Sepsis syndrome/severe sepsis:Continuation and adjustment of antimicrobial therapy based on microbiological findings and clinical response.Enhanced supportive care, including mechanical ventilation for respiratory failure and renal replacement therapy for acute kidney injury.Glycemic control and nutritional support as adjunctive treatments.

Septic shock:Aggressive hemodynamic support with fluids and multiple vasopressors to maintain systemic perfusion.In refractory shock: adjunctive hydrocortisone (200 mg/d).Consideration of inotropic support for myocardial dysfunction.Blood purification techniques: Introduced in this stage for patients with refractory shock and/or significant organ dysfunction, aimed at removing excessively elevated inflammatory mediators and toxins. Blood purification techniques are considered adjunctive therapies and are typically reserved for cases where conventional treatments fail to stabilize the patient or when there is evidence of overwhelming inflammation contributing to organ dysfunction [6]. Their use is guided by the severity of sepsis, the patient’s response to initial treatments, and the presence of complications such as severe metabolic derangements or refractory shock.

It is crucial to note that the efficacy and timing of blood purification techniques in sepsis management are the subject of ongoing research, and their application should be considered within the context of a comprehensive, evidence-based treatment plan and the availability of specialized resources.

### 1.2. Immune Response Mechanisms in Sepsis: From Recognition to Regulation

Both adaptive and innate immunity rely on a variety of intracellular, membrane-bound, and soluble receptors. These include pattern recognition receptors (PRRs) which detect not only pathogen-associated molecular markers (PAMPs, e.g., endo- and exotoxins, DNA, lipids) from foreign invaders but also endogenous, host-derived danger signals (damage-associated molecular patterns, DAMPs). The recognition of PAMPs or DAMPs triggers a cascade aimed at both containing and destroying invasive pathogens, as well as repairing damaged tissue.

The resulting upregulation of pro- and anti-inflammatory signaling pathways leads to a systemic release of cytokines, mediators, and pathogen-related molecules. This, in turn, activates coagulation and complement cascades, contributing to the immune response [7].

Toll-like receptors (TLRs), a subclass of PRRs, are located on the outer membranes as well as in intracellular vesicles of antigen-presenting cells (APCs) and monocytes [8]. Their interaction with PAMPs and DAMPs (e.g., extracellular LPS or intracellular nucleic acids) initiates signal transduction, which triggers a translocation of the nuclear factor-kappa-light-chain-enhancer of activated B cells (NF-κB) into the cell nucleus. This, in turn, leads to the expression of “early activation genes”. These genes include proinflammatory interleukins (IL), such as IL-1, IL-12, IL-18, along with tumor necrosis factor (TNF) and interferons (IFN). These proinflammatory substances then promote the activation of complement and coagulation pathways and stimulate the release of further cytokines (e.g., IL-6, IL-8, IFN-γ). Furthermore, negative feedback mechanisms lead to the downregulation of components of the adaptive immune system [9]. These processes can be observed in the early stages of sepsis and are characterized by a sharp increase in both proinflammatory and anti-inflammatory cytokines [10,11,12].

This excessive and widespread increase in pro- and anti-inflammatory cytokines, resulting from the upregulation of both proinflammatory and anti-inflammatory signaling pathways, is a classic hallmark of sepsis. It leads to progressive tissue damage in the host and can ultimately escalate into multiorgan dysfunction. In the later stages of sepsis, the downregulation of activating cell surface molecules, increased apoptosis of immune cells, and T-cell exhaustion often result in emerging immunosuppression, a phenomenon known as “immune paralysis”. This condition renders affected patients susceptible to nosocomial infections, viral reactivation, and opportunistic pathogens [10,13].

### 1.3. Neutrophils in Sepsis: Roles in Defense, Hyperinflammation, and Organ Damage

The immunological characterization of sepsis is complicated by its highly variable influence on the immunological phenotype, which can manifest itself either as hypo- or hyperreactive, or as a mixed form. Neutrophil granulocytes, a key component of the innate immune system, play a crucial role in the primary defense against pathogens. They contribute to hyperinflammation in sepsis through the release of proteases and reactive oxygen species.

In response to severe bacterial infections, both mature and immature forms of neutrophils are released from the bone marrow in a process known as emergency granulopoiesis. However, when activated by interaction with PAMPs or DAMPs, immature neutrophils show reduced phagocytosis and limited oxidative burst capacity [14,15,16].

Neutrophil granulocytes are capable of releasing neutrophil extracellular traps (NETs) [17]. NETs are diffuse extracellular structures composed of decondensed chromatin with granular and nuclear proteins (histones) that can bind to endothelial or epithelial cells, potentially causing cell damage. This may promote the formation of intravascular thrombi and contribute to multiple organ damage [18,19,20]. NETs are also known for their ability to immobilize a wide range of pathogens. In addition to Gram-positive and Gram-negative bacteria, this includes viruses, yeasts, and even larger organisms, such as protozoa and parasites, which cannot be regularly phagocytosed due to their size [21,22].

In addition to various cytokines like IL-8, IL-1β, and TNF, the release of NETs can also be triggered by platelet agonists such as adenosine diphosphate (ADP), arachidonic acid, collagen, thrombin, and some antibodies [17,18,23,24]. Conversely, inhibiting NET formation has been shown to lead to increased bacteremia and subsequently higher mortality rates in animal models of sepsis [25]. Clinical deterioration is often associated with elevated counts of neutrophil granulocytes, which in turn stimulates the increased production and release of NETs [18,23].

### 1.4. Endothelial Dysfunction and Thromboinflammation in Hyperinflammatory Diseases

The endothelium, along with its protective layer of glycoprotein polysaccharides known as the glycocalyx, plays a significant role in the progression of diseases associated with hyperinflammation. Both are key targets in various mechanisms that perpetuate the inflammatory response [26,27].

In such conditions, endothelial cells may lose their antithrombotic properties. For example, the expression of surface-bound thrombomodulin can be reduced, leading to an increase in tissue factor (TF) expression. This, in combination with leukocytic microparticles and monocytes that also carry TF, triggers the activation of the coagulation cascade [28]. Furthermore, microbes, various cytokines, and components of the complement system can induce an increased expression of TF on endothelial cells, macrophages, and monocytes, contributing to “thromboinflammation” [29].

Feedback mechanisms consequently lead to progressive vascular hyperpermeability, increased recruitment of inflammatory cells, pronounced expression of adhesion molecules, and the release of additional cytokines.

The binding of released TF to activated platelets and neutrophils, among others, further intensifies the prothrombotic situation. Simultaneously, the activity of antithrombotic factors, including antithrombin, the protein C system, and the tissue factor pathway inhibitor (TFPI), is reduced [30].

### 1.5. Complement System Activation and Immunothrombosis in Sepsis and Systemic Inflammation

The complement system is a crucial component of innate immunity. In the initial phase of systemic hyperinflammation, elevated levels of activated complement factors such as the proinflammatory peptide fragments C3a, C4a, and C5a can be detected [24]. These anaphylatoxins, particularly C5a, intensify various responses, ranging from triggering apoptosis to the functional deactivation of neutrophils and the amplification of the hyperinflammatory response. C5a is known for its role in neutrophil chemotaxis; neutrophils, upon binding to the C5a receptor (C5aR), acquire the ability to migrate to and invade inflamed tissues. There, through the binding of PAMPs and DAMPs, they become activated and release granular enzymes, reactive oxygen species, and NETs [31]. Elevated levels of C5a are associated with a worse clinical outcome, due to increased systemic inflammatory response and apoptosis [32].

Evolutionarily, the complement and coagulation systems share a common origin. The release of the proinflammatory complement factors C3a and C5a simultaneously not only leads to the recruitment but also the activation of platelets, endothelial cells, and leukocytes. Coagulation can be activated by coagulation factor XI, or, alternatively, through the cleavage of kininogen with release of bradykinin and antimicrobial peptides. Subsequent research suggests that, under certain conditions, thrombosis may play a significant physiological role in immune defense. Consequently, inhibiting coagulation could also impair antimicrobial defense. This understanding led to the introduction of the term “immunothrombosis” in 2013 [33]. In mammals, well-conserved links between hemostasis and inflammation have been discovered. For example, coagulation factors like Factor IIa or Xa can induce the release of cytokines via activation of protease-activated receptors, potentially contributing to the inflammatory response [34]. In modern understanding, both the activation of the human contact system and intrinsic coagulation (manifesting as coagulopathy) are recognized as part of the innate immune response [35].

Clinically, coagulopathy is a frequent complication of sepsis, and can be detected in up to one-third of critically ill patients. The International Society of Thrombosis and Haemostasis (ISTH) describes disseminated intravascular coagulopathy (DIC) as a syndrome “characterized by the intravascular activation of coagulation with loss of localization arising from different causes. It can originate from and cause damage to the microvasculature, which, if sufficiently severe, can produce organ dysfunction” [36]. The occurrence of DIC in sepsis is attributed to consumptive coagulopathy, driven by system-wide coagulation activation and accompanied by suppressed fibrinolysis. Alongside organ dysfunction due to systemic inflammation, decreased platelets, and increased PT-INR, the term “sepsis-induced coagulopathy (SIC)” has been introduced to describe this condition [37].

To summarize, many of the complex and diverse processes associated with septic hyperinflammation occur in the plasma. The various substances and messengers involved are systemically elevated and are present in a dissolved form, making them potential targets for treatment through blood purification. Despite numerous approaches being explored over the past decades, no single procedure nor combination of techniques has yet been identified that significantly improves the survival rates of patients with sepsis. Figure 1 illustrates a range of different techniques for extracorporeal blood purification, which are examined in this article for their application in treating sepsis and septic shock.

Figure 2 provides a schematic overview of the potential clearance properties of different blood purification methods based on the molecular weight of various mediators and toxins. Due to the large number of different commercially available membranes and technical settings, there is a wide variance in the actual effectiveness of the respective techniques. An overview of recent meta-analyses, reviews and other publications on the extracorporeal blood purification techniques described in this article for use in sepsis or septic shock is provided in Table 1.

## 2. Renal Replacement Therapies (RRTs)

### 2.1. High-Cut-Off Membranes

Unlike standard high-flux membranes, high-cut-off (HCO) membranes feature increased pore size (20 nm instead of 10 nm), which theoretically allows for a more effective elimination of inflammatory mediators. The use of HCO membranes is similar to standard renal replacement therapy with a prescribed dose ranging between 25 and 40 mL/kg/h, as recommended by Kidney Disease Improving Global Outcomes (KDIGO). HCO membranes are employed in sepsis as well as in other conditions, such as acute kidney injury in the context of rhabdomyolysis or cast nephropathy in multiple myeloma. Initial studies on patients with sepsis-induced acute kidney injury predominantly indicated a more effective clearance of proinflammatory cytokines using HCO filters compared to classical high-flux filters. In a clinical trial involving 24 patients with sepsis-induced acute renal failure, Morgera et al. found that while the HCO membrane was effective in removing inflammatory mediators such as IL-1, IL-6, and TNF through convection, it also resulted in significant albumin loss compared to diffusion-based modalities [46]. Another study confirmed the higher sieving coefficient and mass removal rate of ultrafiltration for certain cytokines but failed to demonstrate a reduction in cytokine plasma levels in critically ill patients with acute kidney injury (AKI) within the first 72 h of therapy [47]. Following a small (*n* = 16) retrospective observational study that suggested a positive effect on mortality (37.5% mortality with HCO filter vs. 87.5% with continuous veno-venous hemodiafiltration, *p* = 0.03), these results were to be verified by the randomized High Cut-Off Sepsis Study (HICOSS) [48]. However, this trial was discontinued after a planned interim analysis showed no benefit in 28-day mortality (31% for the HCO group vs. 33% for the conventional group) or reductions in catecholamine use, days on mechanical ventilation, or duration of intensive care unit (ICU) stay. When diffusive modalities were used, albumin levels did not differ significantly. In summary, at present, there is no evidence supporting a positive effect of HCO filters in sepsis beyond established indications such as rhabdomyolysis.

### 2.2. High-Volume Hemofiltration

Continuous hemodialysis or hemodiafiltration with high filtration volume is likely the oldest method for extracorporeal removal of small molecules. Hemofiltration operates through convection, where dissolved substances are transported along with a solvent across a semipermeable membrane (ultrafiltration), driven by a positive transmembrane pressure gradient. The clearance in this process depends on the ultrafiltration rate, the sieving properties of the membrane for the solute, and the molecular size of the solute. According to the consensus definition, high-volume hemofiltration (HVHF) uses a convective target dose of more than 35 mL/kg/h, while a target dose of more than 45 mL/kg/h is classified as very-high-volume hemofiltration (VHVHF) [49]. As these methods do not require additional elements to be added to the standard circuit, they can be readily implemented as long as there is experience in the use of continuous renal replacement therapies. These techniques have been employed for immunomodulation in sepsis by aiming to eliminate inflammatory mediators through convection. Although most inflammatory molecules are medium-molecular substances and, in theory, can be removed by this technique, their endogenous release rate in sepsis is significantly higher compared to uremic toxins. Various studies have investigated the effects of different therapeutic regimens on outcome in sepsis and septic shock, using different target doses (HVHF and VHVHF) as well as comparing intermittent versus continuous usage [50,51,52,53]. Although a meta-analysis indicated lower mortality and improved hemodynamics, characterized by a lower heart rate and higher mean arterial pressure, it did not demonstrate a significant impact on disease severity or oxygenation index. Furthermore, most of the RCTs included in the meta-analysis were not of high quality, leading to questionable reliability of findings for various parameters (e.g., IL-6, mean arterial pressure) [54]. Therefore, the data available to date are insufficient for a conclusive assessment. Future studies should focus on exploring alternative extracorporeal therapies, rather than concentrating solely on HVHF as an adjunctive therapy for sepsis.

## 3. Adsorption

Hemoadsorption is a technique where blood is exposed directly to sorbents in an extracorporeal circuit. Initially, these materials were primarily resins or charcoal. A range of physicochemical interactions, including electrostatic attraction, van der Waals forces, hydrogen bonds, and hydrophobic interactions, lead to nonspecific adsorption of numerous small and medium-sized molecules. By changing the structure of the adsorption materials or employing artificial high-tech polymers, it is possible to specifically enhance the selectivity for certain substances, in addition to improving binding capacity. The earlier issue of poor biocompatibility of the used substances used has been largely addressed with the introduction of biocompatible coatings, among other solutions. Owing to its capacity to adsorb larger molecules, which surpass the molecular weight limit of synthetic high-flux dialysis membranes, hemoadsorption is a potentially suitable technique for the treatment of sepsis.

### 3.1. Polymyxin B-Immobilized Fiber Columns (Specific Hemoadsorption)

In cases of Gram-negative sepsis, endotoxin (lipopolysaccharide (LPS) and its fragments) triggers the activation of different cell types, including endothelial cells, monocytes, polymorphonuclear neutrophils, and tissue-resident cells, as well as plasmatic systems like the complement and coagulation pathways. Endotoxin falls under the category of PAMP, and high serum activity of endotoxin is seemingly associated with increased disease severity and impacts survival rates in patients with sepsis or suspected sepsis [55,56]. That said, developing extracorporeal systems to remove this triggering stimulus from the bloodstream appears logical. One of the most promising approaches in this regard is hemoperfusion with polymyxin B-immobilized fiber columns (PMX). Polymyxin B, a cyclic lipophilic peptide antibiotic, is extensively studied for neutralizing LPS due to its high affinity for the lipid A moiety of endotoxin. This treatment approach was first applied to patients with abdominal sepsis. The device has been evaluated in two RCTs for sepsis or septic shock with an abdominal focus: EUPHAS and ABDO-MIX [57,58,59]. While the EUPHAS study showed a trend towards reduced mortality, this finding could not be confirmed by the ABDO-MIX study. One possible reason for this discrepancy might have been frequent “clotting” of the PMX cartridges, which resulted in only 70% of the cohort completing two treatments of two hours each. Another clinical trial, EUPHRATES, was specifically designed to investigate the impact on mortality in patients with septic shock and high endotoxemia, defined by an endotoxin activity assay (EAA) score of ≥0.6 [60]. The EAA is a semiquantitative assay where endotoxin activity is measured as the relative oxidative burst of primed neutrophils, detected through chemoluminescence. The results are expressed as EAA units on a scale from 0 (no endotoxin) to 1 (maximum burst) [61,62]. Upon completion of the EUPHRATES trial, which enrolled 450 patients, it was found that the primary endpoint of 28-day mortality was not met in the “per-protocol analysis” [63]. However, a subsequent post hoc analysis of the data indicated a significant reduction in mortality and improvements in mean arterial pressure, as well as an increase in ventilator-free days in a subset of patients with endotoxin activity levels between ≥0.6 and 0.89 [64]. The exclusion of patients with an EAA ≥ 0.9 suggests that there might be an upper limit of endotoxin load beyond which PMX treatment is less effective. Based on these findings, the use of PMX remains an intriguing option and should be investigated further [65].

### 3.2. LPS Adsorber

The LPS Adsorber is a commercially available medical device designed for extracorporeal blood purification, specifically targeting the elimination of circulating endotoxin (lipopolysaccharide, LPS) from the bloodstream. This device features a cartridge filled with discs made of porous polyethylene (PE), characterized by surface pores averaging 100 μm in size. These surfaces, along with the pores, are coated with a specially designed peptide, synthesized entirely via solid phase peptide synthesis. This method ensures that the peptide is not genetically engineered and does not originate from human or animal sources.

The peptide, covalently bound to the cartridge, is cationic and exhibits a high affinity for the negatively charged lipid A domain of LPS. Notably, even picomolar levels of lipid A can activate macrophages and stimulate the release of proinflammatory cytokines such as IL-1β and TNF. The LPS Adsorber has an adsorption area of about 4 m^2^, with a surface designed to minimize any systemic reactions upon contact with blood.

Results from the pilot Phase IIa trial were published 2020 in Shock. This trial was aimed to allocate 32 septic shock patients with abdominal or urogenital focus in six Scandinavian ICUs who were randomized to either LPS Adsorber therapy or a Sham device. After 527 days, the investigation was terminated with only 15 patients included (eight in the LPS Adsorber group, seven in the control group). LPS levels in plasma were low without group differences; also, the chances in organ function and inflammatory markers were similar in both groups [41].

### 3.3. CytoSorb^®^ (Unspecific Hemadsorption)

The commercially available CytoSorb^®^ device, which is approved for medical use, employs a nonselective hemadsorption process. It consists of a cartridge filled with beads made of a highly porous resin, coated with biocompatible polyvinylpyrrolidone. Despite these hollow spheres having a diameter of only around 300–600 μm, the active surface area of a cartridge is approximately 45,000 m^2^, significantly surpassing the surface area of conventional hemofilters, which is typically around 1.2–2.5 m^2^. When integrated into in a conventional extracorporeal system, such as continuous renal replacement therapy (CRRT) or extracorporeal membrane oxygenation (ECMO), the patient’s blood is passed over the adsorptive surface of the cartridge. This process facilitates the selective adsorption of various substances and molecules within the range of ~5–60 kDa, depending on their plasma concentration. In addition to substances like free hemoglobin, myoglobin, bilirubin, bile acids, and bacterial toxins (excluding endotoxin), activated complement components, some drugs, cytokines, and inflammatory mediators (such as IL-1b, IL-6, IL-8, IL-10, and TNF) can be absorbed. However, evidence-based data on the use of CytoSorb cytokine absorption are still limited due to the scarcity of randomized clinical trials.

An initial multicenter study conducted in 2013 indicated a reduction in the systemic IL-6 concentration following CytoSorb^®^ application in septic patients. Yet, there was no evidence of a reduction in mortality, and the size of the study (*n* = 43 patients) was not sufficient to determine such outcomes [66].

In a case series involving 26 patients with septic shock and renal replacement therapy, a rapid stabilization of hemodynamic parameters, a reduced need for vasopressors, and a decrease in serum lactate were observed [67]. Additionally, when cytokine adsorption was initiated within 24 h of the onset of sepsis, these patients exhibited a lower observed mortality compared to what was predicted by the APACHE II score. It is important to note, however, that this study did not include a control group, which limits the ability to draw definitive conclusions about the effectiveness of the intervention.

In a prospective monocentric open-label study conducted in 2017 with 20 consecutive patients experiencing refractory septic shock, cytokine adsorption was employed as a rescue therapy [68]. The authors reported a significant reduction in vasopressor requirements and an increase in lactate clearance, resulting in the resolution of septic shock in 13 patients (65%). These results, along with a significant decrease in procalcitonin levels, were later confirmed in a randomized trial [69]. However, while the adsorption capacity for IL-6 was confirmed in a smaller randomized study, it did not demonstrate a decrease in systemic IL-6 levels or a reduction in mortality [70]. Additionally, a retrospective analysis of 67 patients, along with a control cohort selected through propensity score matching, also evaluated data from the use of CytoSorb therapy in septic shock. This analysis indicated a reduction in mortality, using the expected mortality according to the Sequential Organ Failure Assessment (SOFA) score as a reference [71].

Despite the growing body of research data, including numerous individual case reports and case series, there is still insufficient evidence to broadly recommend cytokine adsorption for the treatment of sepsis and septic shock. Ongoing randomized controlled trials are targeting patient groups where hypercytokinemia is documented (e.g., IL-6 above 1000 pg/mL), as this seems more biologically plausible for the treatment’s intended effect. A retrospective evaluation involving more than 500 patients, which incorporated the newly validated “CytoScore”, suggests that CytoSorb should be initiated as early as possible in the course of sepsis or septic shock, particularly in very sick patients [72]. The findings indicated that the earlier the CytoSorb therapy was started, the better the outcome in terms of mortality. This score could potentially be useful for stratification in future studies, providing a more targeted approach to patient selection and treatment timing.

## 4. Therapeutic Plasma Exchange (TPE)

The balance between circulating cells and the vascular endothelium is maintained through the interplay of various proteins and receptors. A key protein in this interaction is von Willebrand factor (vWF), which has a multimeric structure. The equilibrium of vWF is regulated by a disintegrin and metalloproteinase with a thrombospondin type 1 motif, member 13 (ADAMTS13), also known as von Willebrand factor-cleaving protease (vWFCP). A deficiency in ADAMTS13 activity can lead to markedly elevated levels of large vWF multimers, resulting in thrombocytopenic microangiopathy (TMA). Two notable and extreme forms of TMA are thrombotic thrombocytopenic purpura (TTP) and thrombocytopenia-associated multiple organ failure (TAMOF). In cases of thrombotic thrombocytopenic purpura, therapeutic plasma exchange (TPE) is part of the standard therapy. TPE, which is well known in nephrology and hematology, is a comparatively complex procedure. The potential for nondiscriminatory removal of cytokines and mediators of inflammation has led to the exploration of TPE as a therapeutic approach in sepsis and septic shock. TPE not only facilitates the effective elimination of damaging circulating molecules but also enables the replenishment of essential plasma components that are depleted by the disease process. These components include antipermeability factors such as ADAMTS13, angiopoietin-1, and protein C, all of which are abundantly present in fresh frozen plasma (FFP) [73]. A deficiency of ADAMTS13, the pathophysiological correlate of TTP, results in vWF released by the endothelium not being adequately cleaved into smaller fragments. This leads to stasis of the microcirculation, subsequently impairing metabolism in the affected organs [74]. ADAMTS13 levels are reduced in septic shock and this reduction is associated with increased mortality, suggesting that substitution through TPE might be a promising approach [75]. However, data on the use of plasmapheresis in sepsis and septic shock remain limited. A meta-analysis published in 2014 by Rimmer et al. failed to provide evidence that would justify a broad application. In addition to the small number of analyzed RCTs, the patient population consisted of a very heterogeneous group of patients and included a high proportion of pediatric patients [76].

In 2018, Knaup et al. published the results of a prospective, monocentric pilot study in which 20 patients in early septic shock were treated with TPE [77]. The study reported improvements in hemodynamics and a reduced need for catecholamines. To validate these findings, the bicentric EXCHANGE study was conducted with 40 patients; however, this study was not powered to demonstrate a significant difference in organ dysfunction or mortality [78]. The upcoming multicenter EXCHANGE-2 trial aims to address these gaps. This trial is set to begin recruiting a planned 274 patients and will provide further insights into the efficacy of TPE in septic shock. David and Stahl (2019) suggest that plasmapheresis should be considered a viable therapeutic option for suitable patients, particularly due to its capacity to replace consumed protective factors that maintain microcirculatory flow and counteract vascular leakage [73]. A recent meta-analysis was published, analyzing five clinical control trials with a total of 390 patients [79]. The authors used the Newcastle–Ottawa scale (NOS) to ensure the quality of the trials. Although all of the trials had a low risk of bias according to the NOS, a heterogeneous cohort pattern emerged (e.g., one multicenter study alongside four monocentric projects, three studies included exclusively children). Finally, however, the authors concluded that plasmapheresis appears to be a potentially promising treatment option for sepsis patients. However, further randomized and controlled studies are needed to confirm this assumption.

## 5. Combination Methods

### 5.1. oXiris^®^

The oXiris^®^ hemofilter (Baxter, IL, USA) represents a novel approach in the simultaneous removal of inflammatory mediators, endotoxin, fluid, and uremic toxins. This is achieved through the inherent hydrogel structure of the AN69 membrane. The membrane is composed of a three-layer structure and is highly electrically charged. The first layer consists of AN69 copolymer hydrogel structure, in which negatively charged methallyl sulfonate molecules are incorporated, through which cytokines, among others, are adsorbed. In addition, solutes are removed by convection through membrane pores (cut-off 40 kDa). The middle zone consists of polyethyleneimine (PEI), a positively charged multilayer linear structure, which improves biocompatibility and can adsorb negatively charged endotoxin. The third layer, in direct contact with the blood, is coated with heparin, which minimizes local thrombogenicity [80,81,82]. The initial clinical results for the oXiris filter are encouraging, with significant catecholamine savings observed in retrospective studies [83]. In a bicentric randomized controlled trial, the device was used in septic shock and acute kidney failure caused by infection with Gram-negative pathogens, and a reduction in inflammatory cytokines and catecholamine dose was reported [80].

A recent meta-analysis, which compiled clinical data from 10 cohort studies and 4 RCTs encompassing 695 patients with sepsis undergoing CRRT, suggested potentially positive effects on 7-, 14-, and 28-day mortality, SOFA score, catecholamine requirement, lactate level, and length of ICU stay [43]. However, the quality of the studies included was not sufficient to draw definitive conclusions. Again, high-quality RCTs with substantial sample sizes are needed.

### 5.2. Coupled Plasma Filtration Adsorption (CPFA)

CPFA, developed in the 1990s as a treatment for sepsis, involves a two-step process. Initially, plasma is separated from cellular blood components using a highly permeable filter similar to standard plasmapheresis. Then, within the plasma component, adsorption therapy is performed using a styrenic polymer resin before the purified plasma is reinfused back into the patient. This method also allows for simultaneous CRRT for renal support and control of fluid balance. Due to the absence of direct contact between blood cells and sorbent material, CPFA is claimed to have high biocompatibility [84]. However, so far, this innovative approach has not demonstrated a distinct treatment advantage in the limited studies conducted to date. The largest RCT to date (*n* = 192) was terminated early in 2014 for futility, with no evidence of a difference in hospital mortality or ICU-free days (191). The follow-up trials COMPACT-2 and ROMPA were also terminated early in 2017. As the COMPACT-2 trial found significantly increased mortality in the therapy group within the first 72 h of study inclusion, this finding ultimately led to the discontinuation of ROMPA [85,86]. Currently, no other studies are known to investigate the effect of CPFA in sepsis therapy, and it seems unlikely that CPFA will be used in sepsis therapy in the future [44].

## 6. Albumin Dialysis

Conventional dialysis utilizes diffusion, filtration, and osmosis to remove waste products, toxins, and excess fluids from the blood. Yet, this method has limitations in removing larger molecules, such as albumin-bound toxins or inflammatory mediators. Albumin, the most abundant protein in human blood plasma, plays a pivotal role in maintaining colloid osmotic pressure. This pressure primarily arises from the concentration gradient of albumin between the fluid in the blood vessels and the surrounding tissues. Albumin also binds and transports hydrophobic substances in the blood, including certain amino acids, hormones, and fat-soluble substances. Furthermore, albumin has also been recognized for its capacity to bind several inflammatory mediators, exhibiting an immunomodulatory effect in systemic inflammation and sepsis via toll-like receptor-mediated signaling [87,88].

In cases of renal insufficiency, bound molecules may accumulate as they are too large to pass through the pores of conventional dialysis membranes. Albumin dialysis is a highly effective treatment to remove such noncovalently albumin-bound substances utilizing specific semipermeable membranes. In this process, the blood flows along one side of the membrane, while a dialysis fluid containing albumin is present on the opposite side. This “fresh” albumin provides binding capacity for the toxins and other albumin-bound substances and binds them after diffusion through the membrane, thereby being effectively removed from the bloodstream.

In the simplest technical variant of albumin dialysis, known as single-pass albumin dialysis (SPAD), the albumin-containing dialysate is discarded after a single contact with the membrane. Considering the high cost of albumin, the daily therapy costs for SPAD often become a limiting factor in its widespread clinical use [89].

To date, various technologies have been developed where the used albumin is regenerated during ongoing therapy, allowing for multiple uses. This approach conserves resources and reduces costs. Additionally, depending on the specific method, it can increase the effectiveness in removing protein-bound substances and toxins. Examples of such albumin dialysis procedures include the molecular adsorbent recirculating system (MARS^®^), fractional plasma separation and adsorption (FPSA, Prometheus^®^), and the advanced organ support system (ADVOS^®^) [90,91,92]. The techniques employed may include activated carbon adsorbers, ion exchangers, neutral resin adsorbers, or even discharging the albumin through pH-induced conformational changes. Albumin dialysis procedures are primarily used as extracorporeal liver support (ELCS) in patients with acute liver failure (ALF) and acute-on-chronic liver failure (ACLF). Both conditions are associated with high mortality and morbidity, and ELCS procedures are deployed either as emergency therapy or as a bridge to liver transplantation.

A meta-analysis has demonstrated that ECLS may reduce mortality by 16% and may also improve hepatic encephalopathy (HE) in patients with liver failure [93]. However, evidence-based data supporting its use in sepsis are limited and predominantly consist of isolated case reports, particularly in patients without underlying or concomitant liver disease. Consequently, the role of these complex and expensive costly techniques in adjuvant sepsis therapy remains unclear.

## 7. Conclusions

The various techniques for extracorporeal blood purification presented in this article can decrease levels of elevated proinflammatory cytokines in septic shock, potentially mitigating the severity of the systemic inflammatory response. Some methods are effective in removing endotoxins, especially in sepsis caused by Gram-negative bacteria, which may aid in stabilizing the patient’s condition. Reports indicate that blood purification can enhance hemodynamic stability and reduce the need for vasopressors, crucial for managing septic shock. Techniques like CRRT offer simultaneous management of acute kidney injury—a frequent complication in septic shock—alongside the removal of toxins and cytokines.

Many studies have failed to show improved survival rates or significant clinical outcomes, leading to doubts about these treatments’ effectiveness. Blood purification techniques may inadvertently remove beneficial substances, such as essential proteins and immune cells, leading to potential negative consequences. These treatments require specialized equipment and personnel, making them less accessible and more resource-intensive. There is a risk of adverse effects, including bleeding due to anticoagulation and hemodynamic instability during the procedure. The capacity of these techniques to clear mediators like cytokines and endotoxins is often outpaced by their high endogenous production rates in septic shock, which may limit overall effectiveness. The fluctuating levels of cytokines and mediators in the rapidly evolving condition of sepsis make it challenging to target and remove these substances efficiently. The timing and duration of treatment are key; delayed initiation or prolonged treatment might reduce benefits or lead to the removal of beneficial substances and increased complication risks. Blood purification is generally viewed as supportive, assisting in symptom management and stabilization but not addressing the underlying infection or primary sepsis causes. The efficacy of blood purification varies significantly between patients, influenced by factors such as the severity of sepsis, overall health status, and comorbidities. The current evidence base, comprising mostly small or nonrandomized studies, necessitates more high-quality, large-scale RCTs to confirm the efficacy and safety of these techniques.

## Figures and Tables

**Figure 1 ijms-25-03120-f001:**
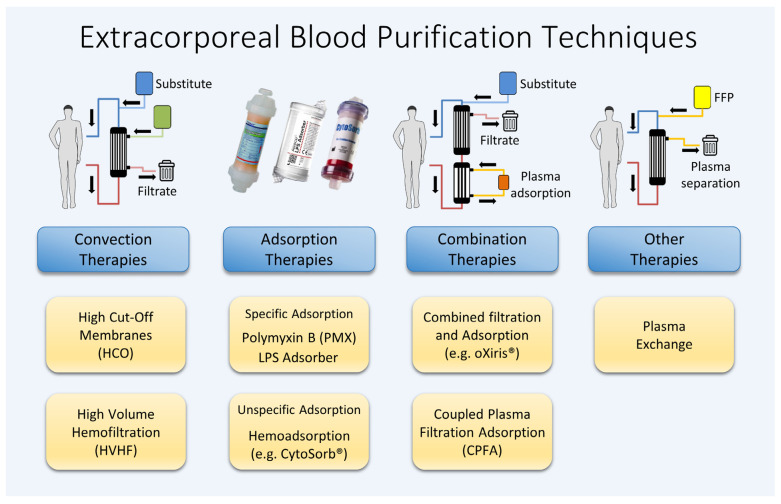
Various extracorporeal blood purification methods available.

**Figure 2 ijms-25-03120-f002:**
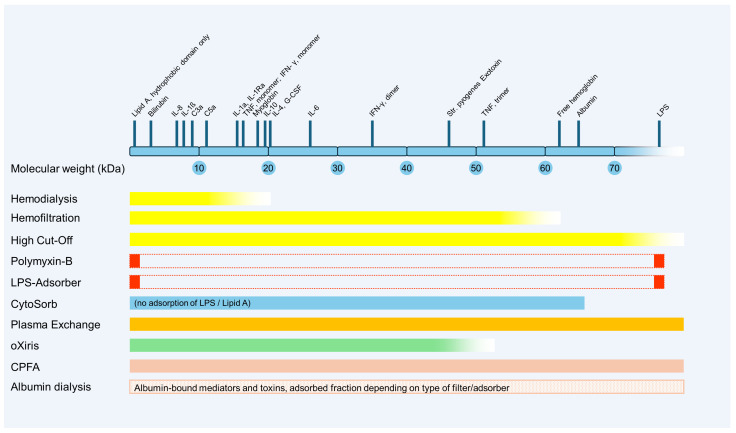
Schematic representation of potential clearance properties of blood purification methods based on the molecular weight of various mediators and toxins. CPFA, coupled plasma filtration adsorption; G-CSF, granulocyte-colony stimulating factor; IFN, interferon; IL, interleukin; kDa, kilodalton; LPS, lipopolysaccharide; TNF, tumor necrosis factor.

**Table 1 ijms-25-03120-t001:** Overview of recent meta-analyses, systematic reviews, and other publications on the described extracorporeal blood purification techniques. CPFA, coupled plasma filtration adsorption; CS, cohort studies; HCO, high cut-off; HVHF, high-volume hemofiltration ICU, intensive care unit; PMX-HP, Polymyxin B-immobilized hemoperfusion; RCT, randomized controlled trial.

Method	Number and Type of Trials	Number of Patients (n)	Patient Status	Results	Reference
HCO	4 RCT 3 OS	215	Sepsis or septic shock	No significant differences in hospital mortality or length of ICU stay. One trial was stopped prematurely for futility after enrolment of 81 patients.	[38]
HVHF Ultrafiltrate rate in intervention group >35 mL/kg/h	5 RCT	241	Sepsis	Available evidence does not support effectiveness in terms of survival. HVHF may be effective in improving individual morbidity.	[39]
PMX-HP	13 RCT	1163	Sepsis or septic shock	Therapy with PMX-HP may reduce mortality, compared to standard of care. Patients with less severe sepsis may benefit more.	[40]
LPS Adsorber	1 RCT	8	Sepsis or septic shock	Terminated prematurely due to recruitment problems.	[41]
CytoSorb	2 RCT 6 CS	776	Sepsis or septic shock	No significant mortality reduction. May be effective in improving ICU morbidity.	[42]
oXiris	4 RCT 10 CS	695	Septic patients undergoing CRRT	Potential association with lower 28-day mortality, decreased norepinephrine dose and shorter ICU stay, no 90-day mortality benefit.	[43]
CPFA	4 RCT 2 CS	537	Sepsis or septic shock	No all-cause mortality benefit.	[44]
Plasma exchange	5 RCT 6 CS	627	Critically ill patients with sepsis-induced multiorgan dysfunction	Potential survival benefit compared to standard of care.	[45]

## Data Availability

Not applicable.

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
