# Peer review of "Septic Hyperinflammation—Is There a Role for Extracorporeal Blood Purification Techniques?"

_ijms, 2024, doi:10.3390/ijms25063120_

Round 1

Reviewer 1 Report

Comments and Suggestions for Authors

This review is well written and provides valuable insights into currently available treatment regimes for septic hyperinflammation.

I have only a few very minor comments to the manuscript that I would  suggest to address before publication:

1) line 72ff: TLR are not only located on the plasma membrane, but also intracellular TLR (on the endosome) - please rephrase this part and take account for the importance of intracellular TLR on IFN release.

2) lines 124 and 127/128: please provide references

3) lines 156-158: please rephrase the sentence as it is not conlusive

4) line 160: please provide examples of "certain coagulation factors"

5) please briefly mention the possible treatment strategies in the various stages of sepsis (only very briefly) to get a better idea of the general options and when the blood purification techniques will be introduced for treatment

6) chapter 3 lines 244ff: please also include the LPS-adsorber here, which uses a specific LPS-binding peptide - there are also clinical trials

7) line 476: please be more specific - instead of "these techniques" (meaning: all of the mentioned techniques or did you only refer to some of the researched ones?)

Author Response

We thank the reviewer for her/his important commenst and suggestions. Below is a point-by-point response to the issues raised by the reviewer:

1) line 72ff: TLR are not only located on the plasma membrane, but also intracellular TLR (on the endosome) - please rephrase this part and take account for the importance of intracellular TLR on IFN release.

Lines 109-112, text added, reference [8] added.

2. lines 124 and 127/128: please provide references

- line 162 – ref. 26 + 27 added.

- line 167 – ref. 28 added.

3. lines 156-158: please rephrase the sentence as it is not conlusive

lines 194-198 – the sentence has been rephrased for clarity

4. line 160: please provide examples of "certain coagulation factors"

lines 199-200 p - we have added examples as requested

5. please briefly mention the possible treatment strategies in the various stages of sepsis (only very briefly) to get a better idea of the general options and when the blood purification techniques will be introduced for treatment

We have added a short paragraph addressing this request (lines 57-94)

6. chapter 3 lines 244ff: please also include the LPS-adsorber here, which uses a specific LPS-binding peptide - there are also clinical trials

Thank you for this important suggestion. We have added a paragraph covering this point (lines 349-369).

7. line 476: please be more specific - instead of "these techniques" (meaning: all of the mentioned techniques or did you only refer to some of the researched ones?)

We have rephrased the sentence for clarity (line 557)

Reviewer 2 Report

Comments and Suggestions for Authors

Thank you very much for this interesting review. The introduction is very extensive and could perhaps be shortened. However it is a good overview. 

The  main body of the text could to my opinion be greatly improved, if there would be created an overview table.  Heading could be: tecnique, number of studies, patients included, results, multocenter or monocentric study.. etc. 

In addtion, it would be great to have a graphical overview about the principle of the different tecniques, 

Author Response

We thank the reviewer for her/his important suggestions. Below is a point-by-point response to the issues raised:

The introduction is very extensive and could perhaps be shortened.

Thank you for this suggestion. Unfortunately, we are unable to fulfill this request since another reviewer suggested to extend the introduction and to address an additional topic.

The  main body of the text could to my opinion be greatly improved, if there would be created an overview table.  Heading could be: tecnique, number of studies, patients included, results, multocenter or monocentric study.. etc. 

- Thank you for this valuable suggestion. We have created a table according to the specifications (table 1, Zeile 238 pp).

In addtion, it would be great to have a graphical overview about the principle of the different tecniques

Thank you. We have created a figure which gives a graphical overview of the points raised (fig. 2, line 232 pp).

Reviewer 3 Report

Comments and Suggestions for Authors
  • The review is devoted to the current problem of sepsis and the possibilities of extracorporeal blood purification to possibly reduce its negative consequences. The first part of the review outlines the mechanisms of development of various immune reactions in sepsis, the task of which is to show the complexity of simultaneously occurring processes. The second part of the review describes the main methods of extracorporeal blood purification and the results of specific clinical studies on their use in sepsis. In general, there are no comments on the review; the only suggestion would be to study a similar review in order to analyze additional studies that could complement this review https://www.reanimatology.com/rmt/article/view/2282/1728

Author Response

We thank the reviewer for her/his important comments. Below is a point-by-point response to the issues raised:

In general, there are no comments on the review; the only suggestion would be to study a similar review in order to analyze additional studies that could complement this review https://www.reanimatology.com/rmt/article/view/2282/1728

We have studied and added the manuscript to the text (line 87) and to the reference section (line 668).